# Towards high-resolution laser ionization spectroscopy of the heaviest elements in supersonic gas jet expansion

R. Ferrer[1], A. Barzakh[2], B. Bastin[3], R. Beerwerth[4,5], M. Block[6,7,8], P. Creemers[1], H. Grawe[6], R. de Groote[1], P. Delahaye[3], X. Fléchard[9], S. Franchoo[10], S. Fritzsche[4,5], L.P. Gaffney[1], L. Ghys[1,11], W. Gins[1], C. Granados[1], R. Heinke[12], L. Hijazi[3], M. Huyse[1], T. Kron[12], Yu. Kudryavtsev[1], M. Laatiaoui[6,7], N. Lecesne[3], M. Loiselet[13], F. Lutton[3], I.D. Moore[14], Y. Martínez[1,15], E. Mogilevskiy[1,16], P. Naubereit[12], J. Piot[3], S. Raeder[1], S. Rothe[15], H. Savajols[3], S. Sels[1], V. Sonnenschein[14], J.-C. Thomas[3], E. Traykov[3], C. Van Beveren[1], P. Van den Bergh[1], P. Van Duppen[1], K. Wendt[12] & A. Zadvornaya[1]

Resonant laser ionization and spectroscopy are widely used techniques at radioactive ion beam facilities to produce pure beams of exotic nuclei and measure the shape, size, spin and electromagnetic multipole moments of these nuclei. However, in such measurements it is difficult to combine a high efficiency with a high spectral resolution. Here we demonstrate the on-line application of atomic laser ionization spectroscopy in a supersonic gas jet, a technique suited for high-precision studies of the ground- and isomeric-state properties of nuclei located at the extremes of stability. The technique is characterized in a measurement on actinium isotopes around the $N = 126$ neutron shell closure. A significant improvement in the spectral resolution by more than one order of magnitude is achieved in these experiments without loss in efficiency.

[1] KU Leuven, Instituut voor Kern-en Stralingsfysica, Celestijnenlaan 200D, B-3001 Leuven, Belgium. [2] Petersburg Nuclear Physics Institute, NRC Kurchatov Institute, 188300 Gatchina, Russia. [3] GANIL, CEA/DRF-CNRS/IN2P3, B.P. 55027, 14076 Caen Cedex 05, France. [4] Helmholtz Institute Jena, Fröbelstieg 3, 07743 Jena, Germany. [5] Theoretisch-Physikalisches Institut, Friedrich-Schiller-Universität Jena, D-07743 Jena, Germany. [6] GSI Helmholtzzentrum für Schwerionenforschung GmbH, 64291 Darmstadt, Germany. [7] Helmholtz Institute Mainz, 55099 Mainz, Germany. [8] Institut für Kernchemie, Johannes Gutenberg-Universität Mainz, Fritz-Strassmann-Weg 2, 55128 Mainz, Germany. [9] Normandie Univ, ENSICAEN, UNICAEN, CNRS/IN2P3, LPC Caen, 14000 Caen, France. [10] Institute de Physique Nucléaire (IPN) d'Orsay, 91406 Orsay Cedex, France. [11] SCK·CEN, Belgian Nuclear Research Center, Boeretang 200, 2400 Mol, Belgium. [12] Institut für Physik, Johannes Gutenberg-Universität Mainz, 55128 Mainz, Germany. [13] Université catholique de Louvain, Centre de Ressources du Cyclotron, B-1348 Louvain-la-Neuve, Belgium. [14] Department of Physics, University of Jyväskylä, PO Box 35 (YFL), Jyväskylä FI-40014, Finland. [15] CERN, CH-1211 Genève, Switzerland. [16] Faculty of Mechanics and Mathematics, Lomonosov Moscow State University, Leninskie gory, 1, 119992 Moscow, Russia. Correspondence and requests for materials should be addressed to R.F. (email: Rafael.Ferrer@kuleuven.be).

The heaviest elements of the periodic table have intriguing atomic and nuclear properties[1]. Valence-electron configurations and ionization potentials are strongly influenced by relativistic effects directly affecting the element's chemical behaviour and thus its position in the periodic table. In heavy nuclei, a competition between the short-range nuclear attraction and the long-range Coulomb repulsion determines their mere existence and properties. Answering questions such as where does Mendeleev's table end and which positions in the periodic table do the heaviest elements occupy need intertwined chemical, atomic and nuclear physics studies.

Element $Z = 118$ is thus far the heaviest element observed and some of its basic decay properties have been determined[2]. Owing to the extremely low production rates, crucial information on nuclear excitation energies, spins and parities, electromagnetic moments and charge radii is scarce for the transactinide isotopes ($Z \geq 104$). Even in the actinide region ($Z \geq 89$) such information only exists for selected, often long-lived isotopes. Copernicium ($Z = 112$) is the heaviest element where chemical properties have firmly been established[3], whereas lawrencium ($Z = 103$) is the heaviest element where the ionization potential is measured[4]. Until recently, information on atomic transitions ended at fermium ($Z = 100$)[5] but has now been extended to nobelium ($Z = 102$), where an optical ground-state transition has been observed[6].

Resonant laser ionization is a well-established atomic physics technique to selectively produce radioactive ion beams and then deduce atomic properties such as ionization potentials, atomic levels, transition strengths, isomer and isotope shifts, and hyperfine constants[7–9]. From these atomic observables, nuclear properties such as spins, magnetic dipole and electric quadrupole moments, and differences in mean-square nuclear charge radii can be deduced in a nuclear-model independent manner. However, current implementation of resonant laser ionization can be limited in the accessibility to all these nuclear observables due to a poor spectral resolution—in the GHz range. Furthermore, the applicability can be hampered for short-lived nuclei, even for certain elements, due to physico-chemical properties.

In this study we describe an approach that is efficient, chemically independent and applicable to short-lived isotopes ($T_{1/2} > 0.1$ s). This major expansion of the so-called in-gas laser ionization and spectroscopy (IGLIS) technique[10,11] has been first examined offline on stable $^{63}$Cu isotopes[12]. We have validated this technique on-line in the actinide region by performing high-resolution laser-spectroscopy studies on the isotopes $^{214}$Ac ($T_{1/2} = 8.2$ s) and $^{215}$Ac ($T_{1/2} = 0.17$ s). Its projected spectral resolution for heavy elements of $\sim 100$ MHz fulfils the requirement for accurate extraction of their ground-state nuclear properties. Moreover, this in-gas-jet laser ionization and spectroscopy method enables the production of extremely pure ion beams, including isomeric beams, which are suitable for further studies.

By choosing the neutron-deficient isotopes of actinium, the first and name-giving element of the actinide group, as a case study, we addressed all challenges for high-resolution resonance ionization spectroscopy of the heavy elements as follows: limited information on the atomic levels due to the absence of stable isotopes, limited production rates due to the necessity of using heavy-ion-induced fusion reactions on thin targets, large background from stronger reaction channels and the need to slow down the energetic radioactive ion beam and transfer it into a controlled ensemble of cooled atoms in a low-density environment, to minimize Doppler and collisional broadening.

Next to the characterization of the technique, we obtained nuclear spins, magnetic dipole and electric quadrupole moments, and differences in mean-square charge radii of neutron-deficient actinium isotopes around the $N = 126$ shell closure by combining the spectroscopic data with atomic calculations. A comparison to large-scale nuclear shell-model calculations provides evidence for the extent of magicity in this region.

## Results

**Production and detection of actinium isotopes.** We produced neutron-deficient actinium isotopes in a complete fusion reaction of $^{20}$Ne or $^{22}$Ne projectiles on a $^{197}$Au target at the Leuven Isotope Separator On Line facility[13] coupled to the CYCLONE accelerator of the Centre de Ressources du Cyclotron (Louvain-la-Neuve, Belgium). The reaction products were thermalized in a gas cell (see Fig. 1), filled with 350 mbar purified argon and were evacuated together with the buffer gas through a

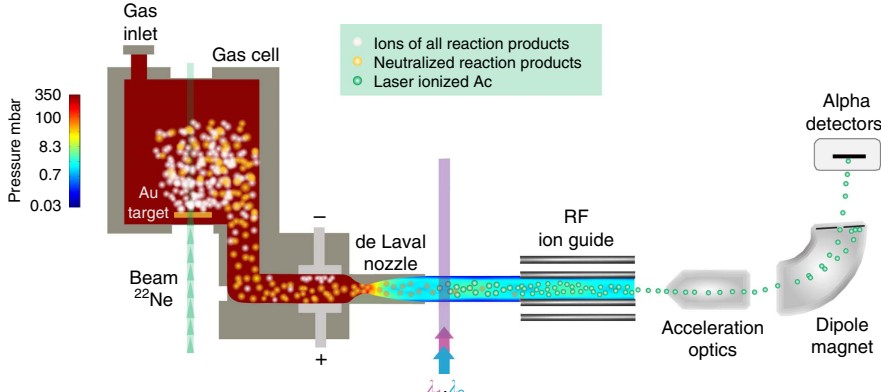

**Figure 1 | In-gas-jet laser ionization and spectroscopy setup.** Short-lived actinium isotopes are produced in the fusion reaction of accelerated neon ions on a gold target. After thermalization and neutralization in purified argon, the actinium atoms are evacuated out of the gas cell through a de Laval nozzle. Just before the nozzle, an electric field is created to collect the remaining ions by applying a DC voltage on a pair of electrodes indicated by a $+$ and a $-$ sign. The resulting collimated supersonic gas jet at Mach $\sim 6$ provides a quasi-collisional free environment at a low temperature ($T \sim 30$ K). The gas jet, containing the reaction products, is overlapped with the laser beams to resonantly ionize actinium. The ions are subsequently sent out from the gas cell chamber towards the mass separator through a radiofrequency (RF) ion guide and their decay radiation is finally recorded. The pressure conditions (colour code on a logarithmic scale) range from 350 mbar in the gas cell to the 0.03 mbar background pressure in the gas cell chamber. The setup is not shown to scale.

de Laval (convergent–divergent) nozzle resulting in a collimated supersonic gas jet. In contrast to former in-gas-cell laser-spectroscopy studies[10,11], the laser beams were now overlapped with the gas jet outside the gas cell, leading to the resonant ionization of actinium in the low-density, low-temperature supersonic gas-jet expansion (see Methods section). The photoions were then captured and transported in a radiofrequency ion guide, accelerated, mass separated and subsequently implanted into an α-decay detection setup. By recording the α-decay spectra as a function of the laser wavelength of the excitation step laser, spectra of the hyperfine structure (hfs) were obtained.

**Determination of atomic transitions in actinium.** Given the lack of information on atomic transitions in actinium, we first performed offline laser-spectroscopic studies using the long-lived isotope $^{227}$Ac ($T_{1/2} = 21.8$ y) to identify an efficient two-step ionization scheme featuring suitable properties for high-resolution spectroscopy studies, that is, a measurable hyperfine splitting and sensitivity to changes in the charge radius (see Methods). As a result of these experiments, we selected the ionization scheme shown in Fig. 2a comprising six multiplets (numbered from I to VI) as the best choice to perform the on-line studies. It is noteworthy that the angular momentum for the level at 22,801 cm$^{-1}$ was identified in these measurements to be $J = 5/2$, which is in contrast to the reported literature value[14]. The assignment of the atomic term as $^4P^o_{5/2}$ is supported by Multi Configuration Dirac Hartree Fock (MCDHF) calculations (see Methods).

**Study of $^{212-215}$Ac in the gas cell.** To identify the corresponding hfs of short-lived $^{212-215}$Ac, we then used the in-gas-cell method[10,11] to perform a broad-band spectroscopy search, which enabled us to narrow down the scanning ranges for the high-resolution spectroscopy study. An example of such a scan for $^{215}$Ac is shown in Fig. 2b. The spectral linewidth of 5.8(2) GHz (full width at half maximum (FWHM)) mainly resulted from collisional and temperature-associated (Doppler) broadening and made it possible to scan efficiently the full ~80 GHz hfs of the $^4P^o_{5/2}$ excited state. However, such a linewidth completely masked the ground-state hyperfine splitting, essential for the spin and quadrupole moment determination.

**Study of $^{214,215}$Ac in the gas jet.** Finally, low-temperature and low-pressure conditions were obtained in the supersonic jet, where laser ionization and spectroscopy studies on the $^{214,215}$Ac isotopes were performed. The hfs of $^{215}$Ac obtained in the jet experiments is compared with the in-gas-cell results in Fig. 2b, whereas the parameters and performances of both approaches are summarized in the second and third column in Table 1. The reduction of both collisional and temperature-associated broadening in the jet experiments led to a 15-fold improvement of the spectral resolution, which enabled a precise fitting of the ground-state hfs, as shown in Fig. 2c for the triplet (IV) and Fig. 2d for the doublet (V) transitions in $^{214}$Ac and $^{215}$Ac, respectively. Owing to technical constraints, the temporal overlap of the laser beams with the atoms in the jet at the given laser repetition rate of 10 kHz (ref. 15) could not be optimized and only

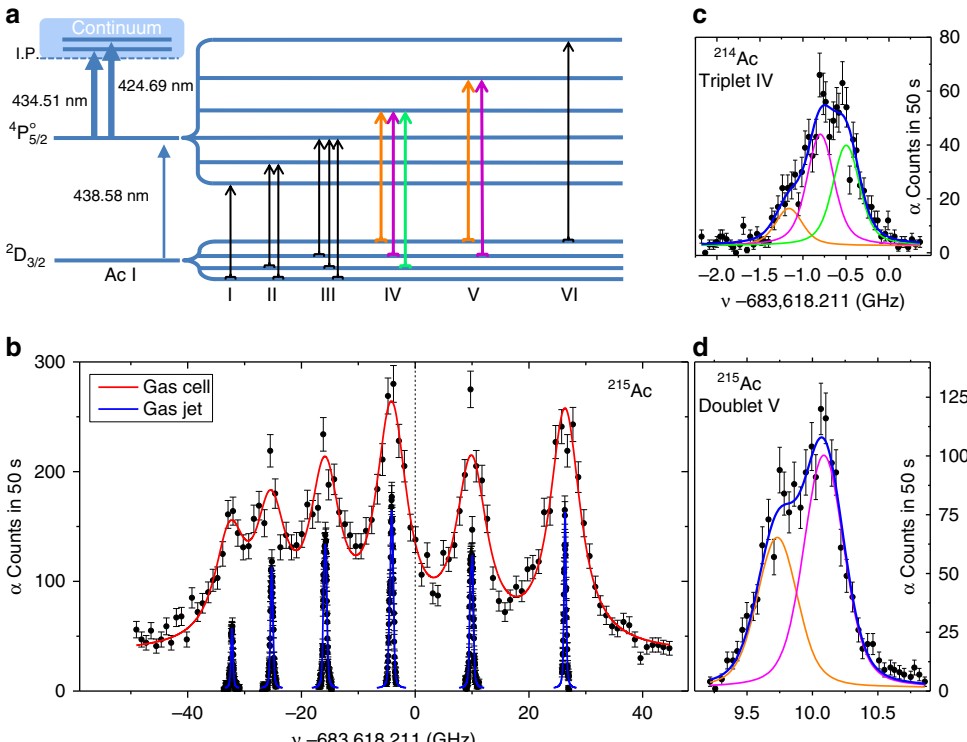

**Figure 2 | Gas cell versus gas jet spectra.** (a) Ionization scheme with vacuum wavelengths for the excitation and ionization steps along with the expected hyperfine splitting for the $^{212-215}$Ac isotopes indicating (not to scale) the 12 hyperfine transitions grouped in six (I to VI) multiplets. (b) Measured spectra (black dots) of the hfs in the $6d\,7s^2\,^2D_{3/2} \rightarrow 6d\,7s\,7p\,^4P^o_{5/2}$ transition from the ground state shown as the α-counts collected in 50 s versus the frequency detuning $\nu$ with respect to the value of the centre of gravity. The gas-cell data has been corrected for the pressure shift (see Methods section). The red (blue) curve shows the best fit of a 12-peak Voigt profile to the data from laser spectroscopy studies in the gas cell (gas jet). (c,d) Zoom in of the triplet (number IV) and doublet (number V) hyperfine transitions in $^{214}$Ac and $^{215}$Ac, respectively, for the typical average energy per pulse of 0.8 μJ of the excitation laser radiation. The colour code used to show the different components of the multiplets indicates the corresponding hyperfine transitions as seen in the excitation scheme in **a**. One sigma s.d. as statistical errors are reported in all the data points.

a fraction of 1/14 of the available atoms was irradiated (see Methods). Even with this small duty cycle, the efficiency was similar to that in the gas-cell studies.

**Ground-state properties of actinium.** The magnetic-dipole-interaction hfs constant $a$, the electric-quadrupole-interaction hfs constant $b$ and the isotope shift $\delta \nu^{A,215}$ were extracted from the obtained spectra and are given in Table 2. The experimental systematic uncertainties originate from the instability and nonlinearity of the different wavelength meters used in the three experiments and from the pressure dependence in case of the in-gas-cell data. Out of these isotope-dependent atomic observables, we could deduce a number of nuclear ground-state properties for the different isotopes, also given in Table 2. The spectral resolution obtained in the offline and in-gas-jet measurements, made it possible to unambiguously determine the nuclear spins $I$ of the isotopes investigated. This results in $I = 3/2$

for $^{227}$Ac and confirms the 1951 measurement[16], $I = 9/2$ for $^{215}$Ac substantiating the previous assignment based on the observation of favoured $\alpha$-decay to $^{211}$Fr (http://www.nndc.bnl.gov/ensdf/) and a spin value of $I = 5$ for $^{214}$Ac, for which no previous assignment is given in the National Nuclear Data Center data base. The in-gas-cell data for $^{212,213}$Ac does not enable a firm spin assignment. A spin $I = (7)$ for $^{212}$Ac is tentatively proposed based on the result from the additivity rule and shell-model calculation, which places the $I = 7^+$ as the ground state $\sim 200$ keV below the $I = 5^+$ and $6^+$ states (see Methods). Based on the observed experimental trend in the magnetic moments of $^{217,215,213}$Ac (see Fig. 3a) a spin $I = (9/2)$ can tentatively be assigned to $^{213}$Ac.

To obtain the magnetic dipole moment $\mu$ or the spectroscopic-quadrupole moment $Q$ from the measured hfs constants, $a$ and $b$, through the equations $a = \mu \cdot B_0 / (I \cdot J)$ or $b = eQ \cdot V_{zz}$, the magnetic field $B_0$ and the electric field gradient $V_{zz}$ created by the atomic electrons at the site of the nucleus must be known. As $B_0$ and $V_{zz}$ depend only on the electronic configuration and not on the specific isotope, one can usually circumvent this by means of a scaling relation with experimentally measured hyperfine-splitting constants and the independently measured magnetic dipole or electric quadrupole moment of a known isotope. However, the only ground-state dipole- and quadrupole-moments available in the literature are, as quoted by the authors, the preliminary values $\mu = 1.1(1)$ $\mu_N$ and $Q = 1.7(2)$ $eb$ for $^{227}$Ac (ref. 17) (it is noteworthy that these values from 1955 to 58 stem from hfs measurements and atomic calculations rather than from independent $\mu$ and $Q$ measurements) and the magnetic dipole moment for the very short-lived ($T_{1/2} = 69$ ns) isotope $^{217}$Ac, inaccessible to optical spectroscopy measurements. Therefore, we performed advanced calculations of the $a/\mu$ and $b/Q$ values (see Methods) for several atomic levels applying the MCDHF method as implemented in the GRASP2k programme[18] and previously applied to other nuclei[19,20].

These calculations, combined with the $a$ and $b$ constants of the corresponding levels measured in $^{227}$Ac, resulted in a magnetic dipole moment $\mu = 1.07(18)$ $\mu_N$ and a spectroscopic electric-quadrupole moment $Q = 1.74(10)$ $eb$ for the $^{227}$Ac ground state. The uncertainty on these values reflects the s.d. of the results from the eight transitions used in the evaluation. Although the obtained values are in agreement with the literature values and have similar error bars, they are more reliable as they are based on superior experimental input and advanced atomic model calculations. Using these values as reference, the magnetic dipole

---

**Table 1 | Actual and expected performance of IGLIS on $^{215}$Ac.**

|  | Gas cell | Gas jet (this work)* | Gas jet (projected)† |
|---|---|---|---|
| *Ionization volume* |  |  |  |
| Pressure (mbar) | 350 (15) | 0.7–1 | $\sim 0.05$ |
| Temperature (K) | 350 (25) | 25–30 | $\sim 9$ |
| Jet divergence (deg.) | — | 10–11 | $< 1$ |
| *Linewidth (FWHM)* |  |  |  |
| Total (MHz) | 5,800 (300) | 394 (18) | $\sim 100$ |
| Lorentz‡ (MHz) | 4,000 (400) | 42 (6) | $< 10$ |
| Gauss§ (MHz) | 1,400 (100) | 280 (30) | $\sim 100$ |
| Selectivity‖ | 8.3 (17) | 121 (27) | $> 3,000$ |
| Efficiency¶ (%) | 0.42 (13) | 0.40 (13) | $> 10$ |

One s.d. uncertainties in the reported values are given between parentheses.
*Information based on experimental data, except for the ionization volume parameters that are deduced from known equations[12] for a nozzle, characterized in fluid-dynamics simulations at Mach number $\sim 6$.
†Predictions and extrapolations of the gas-jet results for working conditions at Mach 10.
‡Singled-out contribution due to gas collisions and the natural linewidth ($\sim 4$ MHz). We obtained the latter from MCDHF calculations of the atomic transition rate ($A = 2 \times 10^{-5}$ s$^{-1}$).
§Singled-out contribution from gas temperature and laser linewidth, and in case of the gas jet data also the jet divergence.
‖Ratio between the $^{215}$Ac ion production with lasers on- and off-resonance.
¶Ratio between the $^{215}$Ac ions entering the mass separator to the $^{215}$Ac nuclei stopped in the buffer gas.

---

**Table 2 | Measured and deduced atomic and nuclear properties.**

| Isotope | $a$ ($^4P^o_{5/2}$) (MHz) | $b$ ($^2D_{3/2}$) (MHz) | $\delta \nu^{A,215}$ (MHz)* | $I$ | $\mu$ ($\mu_N$)† | $Q$ (eb)† | $\delta \langle r^2 \rangle^{A,215}$ (fm$^2$)‡ |
|---|---|---|---|---|---|---|---|
| *Offline* |  |  |  |  |  |  |  |
| $^{227}$Ac$_{138}$ | 2,104.8 (10) (20) | 597 (4) (5) | $-5,8319$ (19) (133) | 3/2 | 1.07 | 1.74 | 1.5101 (5) (39) |
| *In-gas-jet* |  |  |  |  |  |  |  |
| $^{215}$Ac$_{126}$ | 2,377.0 (10) (40) | 13 (26) (20) | 0 | 9/2 | 3.625 (2) (7) | 0.04 (8) (6) | 0 |
| $^{214}$Ac$_{125}$ | 2,498.4 (10) (40) | 48 (22) (20) | 2,969 (14) (40) | 5 | 4.234 (3) (8) | 0.14(6) (6) | $-0.0770$ (4) (10) |
| *In-gas-cell* |  |  |  |  |  |  |  |
| $^{215}$Ac$_{126}$ | 2,386 (17) (90) |  | 0 | (9/2) | 3.64 (3) (14) |  | 0 |
| $^{214}$Ac$_{125}$ | 2,525 (22) (90) |  | 3,008 (170) (390) | (5) | 4.28 (4) (15) |  | $-0.078$ (4) (10) |
| $^{213}$Ac$_{124}$ | 2,385 (31) (90) |  | 4,282 (240) (325) | (9/2) | 3.64 (5) (14) |  | $-0.111$ (6) (8) |
| $^{212}$Ac$_{123}$ | 1,837 (22) (90) |  | 7,710 (270) (285) | (7) | 4.36 (5) (21) |  | $-0.200$ (7) (7) |

Summary of the main results obtained for the actinium isotopes in the different experiments. Statistical ($2\sigma$) and systematic ($1\sigma$) uncertainties, respectively, are given between parentheses.
*The centre of gravity of $^{215}$Ac 683,618,211(12)(100) MHz is used as a reference.
†The $\mu$ and $Q$ values of $^{227}$Ac as deduced from our experimental data and MCDHF calculations (see Methods) are used as reference and have associated uncertainties of 17 and 6%, respectively. These uncertainties are not included in the error balance of the values of the $^{212-215}$Ac isotopes.
‡Values deduced using the calculated isotope shift parameters $M = 500(180)$ GHz amu and $F = -39(2)$ GHz fm$^{-2}$.

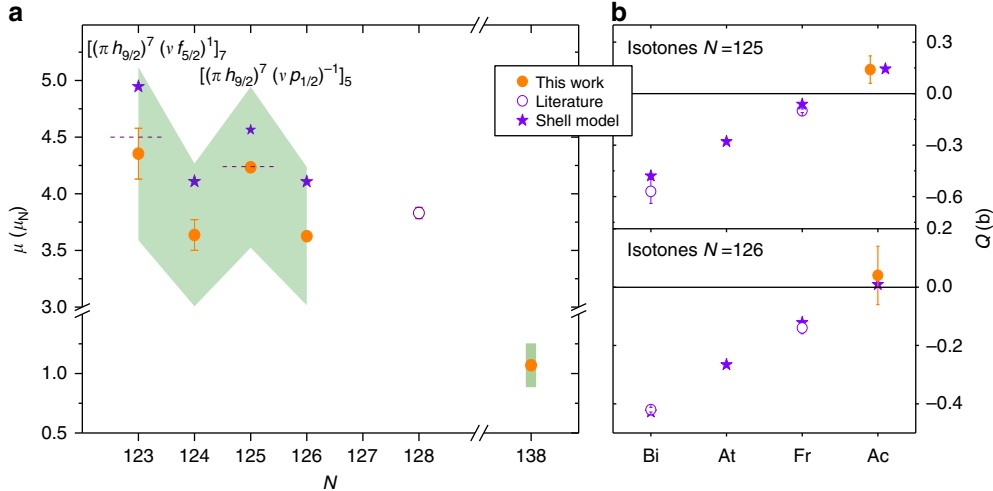

**Figure 3 | Magnetic dipole and electric quadrupole moments. (a)** Experimental magnetic moments of the Ac isotopes (filled dots) compared with values obtained from large-scale shell-model calculations (stars) and from the literature (open dot). Error bars accounting for statistical ($2\sigma$) and systematic ($1\sigma$) s.d. are assigned to our data points, while the shaded band represents the 17% systematic uncertainty from the atomic-physics calculations. The values obtained after applying the additivity rule (dashed lines) using the indicated spin and nucleon configuration are also shown for the odd–odd isotopes. **(b)** The quadrupole moments for the $N=125$ (top) and $N=126$ (bottom) isotones including the actinium isotopes (filled dots) and the neighbouring odd-$Z$ isotopes of francium, astatine and bismuth taken from the literature (open dots) are shown along with the values from shell-model calculations (stars). The literature values are taken from ref. 24 with the exception of those for $^{208,209}$Bi (ref. 49).

moments were deduced for $^{212-215}$Ac from the in-gas-cell measurements and for $^{214,215}$Ac, with a higher precision, from the in-gas-jet measurements. Only with the in-gas-jet method could the electric-quadrupole moments for $^{214,215}$Ac be obtained. The statistical and systematic errors for $\mu$ and $Q$ quoted in Table 2 arise from the uncertainties in the experimental $a$ and $b$ constants. The errors from the $^{227}$Ac reference values are not propagated, leaving the possibility for later improvements of the obtained results once the moments of $^{227}$Ac can be measured in an independent way or the precision of atomic model calculations is improved.

In Fig. 3a our experimentally obtained magnetic dipole moments are shown together with the known ground-state magnetic moment of $^{217}$Ac (ref. 21). We compare these results with shell-model calculations involving $^{208}$Pb ($N=126$, $Z=82$) as core nucleus. For the odd–odd $^{212,214}$Ac isotopes, the results are also compared with values from the additivity relation using the magnetic moment of $^{211,213}$Ra (refs 22,23), respectively, and the magnetic moment of $^{215}$Ac. Both calculations are further explained in the Methods section. The contribution in our data points of the 17% systematic error on the calculated magnetic moment for the $^{227}$Ac ground state is shown with a shaded band. Taking this error into account, our experimental values for $^{212-215}$Ac, which are in good agreement with the results from the additivity relation, do overlap with the shell-model calculations but are systematically lower by $\sim 13\%$. A similar discrepancy is present when comparing our $g$-factor of the $^{215}$Ac ground state with the experimentally known $g$-factors of all $h_{9/2}^{n}$-based states in the $N=126$ isotones[24], with $n$ representing the number of protons occupying the $h_{9/2}$ proton shell ($n_{max}=10$). For pure states, it is expected from angular momentum algebra that $g(h_{9/2})=g(h_{9/2})^{n}$ (ref. 25). For example, in $^{213}$Fr the $g$-factors of the $9/2^-$, $17/2^-$ and $21/2^-$ states are within experimental uncertainty identical[24], whereas our value of $g(9/2^-,^{215}\text{Ac})=0.8056(16)$ differs considerably from the $g$-factors of the $I=17/2^-$ and $21/2^-$ high-spin isomers ($g(21/2^-,^{215}\text{Ac})=g(17/2^-,^{215}\text{Ac})=0.920(19))[24]$. Further developments on atomic theory or an independent measurement of the magnetic dipole moment in $^{227}$Ac are needed to clarify this discrepancy.

As shown in Fig. 3b, the experimental electric-quadrupole moments of $^{214,215}$Ac and literature values for the moments of the neighbouring odd-$Z$ isotones agree with shell-model calculations. The linear increase of the quadrupole moments for $N=126$ and 125 with increasing proton occupation demonstrates a $\pi h_{9/2}^{n}$ seniority-type dependence[26] and for $N=125$ its coupling to a non-contributing $\nu p_{1/2}$ hole (see Methods). The validation of seniority as good quantum label is indicating a robust $N=126$ gap up to $Z=89$. A significant weakening of the $N=126$ shell closure in uranium ($Z=92$) is inferred from a recent analysis of the reduced $\alpha$ widths in the $^{221,222}$U isotopes[27]. No effect is observed in the present data even though increased cross-shell interactions would have a dramatic impact on the small quadrupole moments at $\pi h_{9/2}$ mid-shell ($n=5$).

The centre of gravity of the hfs extracted from the fits for $^{212-214,227}$Ac, relative to that for $^{215}$Ac ($N=126$) defines the isotope shift $\delta\nu^{A,215}$. The isotope shift originates from the change in nuclear mass ($A$) and nuclear charge distribution. From the isotope shift, the difference in mean-square charge radius $\delta\langle r^2 \rangle^{A,215}$ of the different isotopes can be deduced using the calculated parameters $M$ and $F$ (see Table 2 and Methods section). Our measured values exhibit a similar behaviour as those in lighter neighbouring isotopic chains with $Z=82-88$ (refs 9,28). This will be discussed in detail in a forthcoming publication.

## Discussion

Spatial constraints and limitations of the pumping system in the present setup prevented a high-quality jet formation and, as a consequence, an optimal laser-atom overlap in space and time. These will be overcome in future experiments when dedicated IGLIS setups are in operation at new-generation radioactive beam facilities (see for example, ref. 29). Table 1 compares the presently obtained in-gas-jet performance (third column) with the projected performance of the technique under optimal conditions (fourth column), which also include a new gas-cell design with better transport and extraction characteristics[30]. Thus, assuming a duty cycle adapted to the supersonic expansion through a de

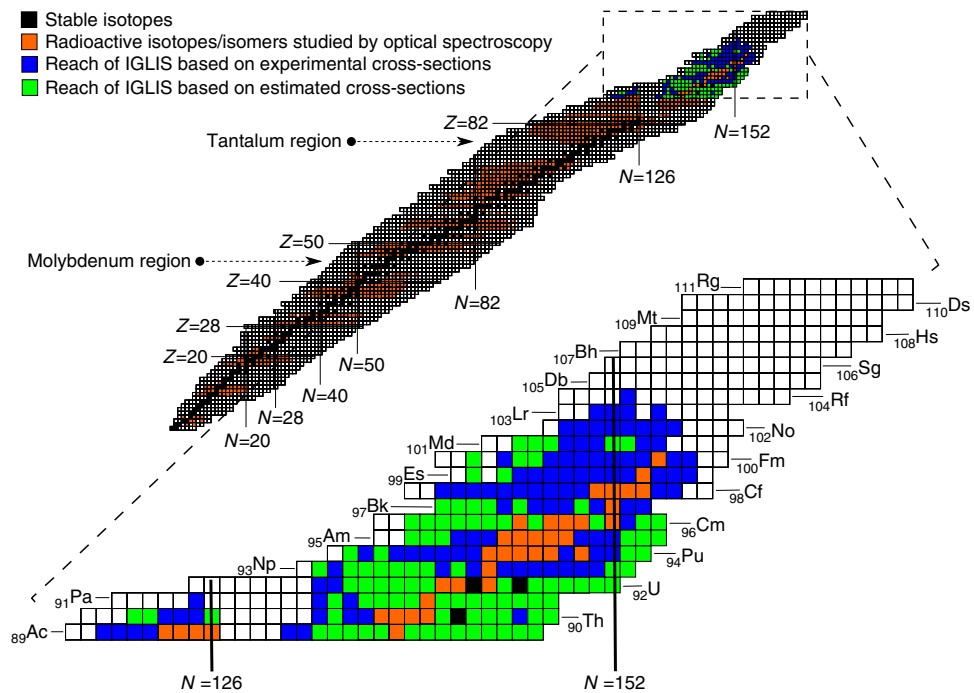

**Figure 4 | Reach of IGLIS for the heaviest elements.** Chart of nuclides showing the current status of the isotopes investigated by optical spectroscopy[9]. Black squares represent stable or very long-lived isotopes, orange squares indicate the radioactive isotopes/isomers with published spectroscopic information, including those from the present work. In the blow-up of the actinide and heavier mass region, blue (green) squares are isotopes that, based on experimental (calculated) cross-sections, can be produced in sufficient amounts to be studied by the IGLIS technique. A primary beam intensity of 10 pμA, the projected efficiency of 10% and yields of up to 0.1 p.p.s. being stopped in the gas cell are assumed.

Laval nozzle with a modest Mach number 10, tuning the background pressure for a pencil-like jet and optimizing the laser interaction zone through a multi-pass mirror system, a final spectral resolution of $\delta\nu/\nu \sim 1 \times 10^{-7}$ ($\sim$100 MHz FWHM) and an overall efficiency beyond 10% can be obtained.

Combined with the fast and chemically insensitive stopping and transport using a purified noble gas, it becomes possible to measure nuclear properties such as charge radii, electric and magnetic moments, and nuclear spins of all isotopes with half-lives above 100 ms and produced at a rate of only 1 atom every 10 s. As the technique is based on laser spectroscopy, fundamental atomic properties such as ionization potentials, transition energies and rates can also be determined. The advantages of this, now proven, approach are obvious when compared with the low-resolution in-source[7,31] or in-gas-cell[8,10] approach but are also evident when compared with the collinear laser spectroscopy technique[9,32], which has a superior spectral resolution but is ultimately limited by the production mechanism of the desired radioactive ion beam and needs at least 100 ions per second.

All radioactive isotopes with published optical data are shown with orange squares on the nuclear chart in Fig. 4. One notices a rather sudden dearth of radionuclides which have been probed as one enters the actinide region and beyond. This echoes a number of significant obstacles in accessing the heaviest of elements including low production cross-sections, strongly competing reaction channels and limited atomic information mainly due to a lack of stable or long-lived isotopes. The reach of the IGLIS technique, based on the projected performance of the in-gas-jet method (see Table 1) and on experimental and estimated cross sections (blue and green squares, respectively), can be seen in the inset of Fig. 4 by the number of new isotopes that can be studied. Furthermore, in the medium-mass region the in-gas-jet laser ionization and spectroscopy method can also be used to study those nuclei, which are inaccessible to conventional

laser-spectroscopy techniques, owing to the physico-chemical properties of the involved elements, such as the refractory elements around molybdenum and tantalum.

In conclusion, the feasibility and impact of the in-gas-jet laser ionization and spectroscopy method have been demonstrated on-line by measuring nuclear and atomic properties of the short-lived isotopes $^{214}$Ac and $^{215}$Ac. The resulting magnetic and quadrupole moments are compared with shell-model calculations and witness a stabilising effect of the $N = 126$ shell up to the actinium isotopes. The obtained efficiency and spectral resolution demonstrate that basic ground- and isomeric-state nuclear properties of heavier actinides and eventually super-heavy elements, as well as their atomic properties, can be determined to high precision. The technique presented has also far reaching consequences for the exploration of the refractory elements, so far hardly accessible to high-resolution laser spectroscopy techniques. In addition, the highly selective ionization enables the production of high-quality, high-purity isotopic and isomeric radioactive ion beams that can be used for other applications in nuclear physics, chemistry and astrophysics, as well as in atomic physics.

## Methods

**Experimental procedure**. To obtain an optimal ionization scheme, we performed offline experiments at the University of Mainz using a number of samples, each containing about $10^{11}$ atoms, of the $T_{1/2} = 21.8$ y isotope $^{227}$Ac. An atomic vapour of $^{227}$Ac was produced by resistively heating the samples in a hot cavity and then step-wise ionized by pulsed Ti:sapphire lasers[33] resulting in an overall efficiency of $10^{-3}$%. We chose the transition from the ground state $6d\,7s^2\;^2D_{3/2} \rightarrow 6d\,7s\,7p\;^4P^{\circ}_{5/2}$ at 438.58 nm for the first-step excitation in combination with two transitions to autoionising states at 434.51 nm[34] and 424.69 nm for subsequent ionization. We characterized in detail the hfs of eight transitions of actinium including that at 438.58 nm[35].

The short-lived $^{212,213}$Ac and $^{214,215}$Ac isotopes were produced at the Leuven Isotope Separator On Line facility in the fusion reaction of 0.16 pμA $^{20}$Ne and $^{22}$Ne projectiles, respectively, impinging with a total energy of $\sim$105 MeV on a $^{197}$Au

target of 1.931 mg cm$^{-2}$ thickness. For the determination of the efficiency of the in-gas-cell and in-gas-jet technique, the experimental cross-sections from refs 29,36 were used. The change in cross-section as a function of the energy loss of the neon beam in the target was taken into account to calculate the production rate. More information on the preparation of the experiments can be found in ref. 15.

We performed low-resolution laser spectroscopy in the gas cell employing a dual-chamber gas cell[37] and a tunable excimer-pumped dye-laser system at a maximum repetition rate of 200 Hz (ref. 13). To characterize the frequency shift due to atom–atom interactions in the high-pressure gas-cell environment, we scanned the singlet hyperfine transition in $^{215}$Ac (number VI in Fig. 2a) for different stagnation pressures. From a linear fit of the centroids and the widths we obtained a collisional shift coefficient $\gamma_{sh} = -3.7(9)$ MHz mbar$^{-1}$, that was used to deduce the position of the hyperfine peaks in the absence of collisions, and a collisional broadening coefficient $\gamma_{coll} = 11.5(10)$ MHz mbar$^{-1}$, leading to a Lorentz contribution in the total linewidth of 4,000(400) MHz (see Table 2). The Gaussian linewidth contribution reported in Table 2 was calculated from the gas temperature and the laser linewidth (1.2(1) GHz).

To realize high-resolution laser spectroscopy in the quasi-collisional free and cold atomic jet we installed at the gas-cell exit a convergent–divergent nozzle designed for Mach number $\sim 6$ with a throat diameter of 1 mm. Under optimal conditions, that is, with a matching of the jet and the background pressures, reached at the dedicated offline laboratory in KU Leuven[30], this leads to a gas jet of about 3 mm diameter, quasi-parallel over >200 mm. In the on-line experiment, where the background pressure could not be controlled sufficiently and where a radiofrequency ion guide was placed at 11 mm distance from the exit of the nozzle, the optimal conditions could not be realized. The supersonic jet, moving with a velocity $v \sim 550$ m s$^{-1}$, was irradiated at normal incidence by a laser system, similar to that employed during the offline studies of $^{227}$Ac, comprising a narrow bandwidth (10–20 MHz) injection-locked Ti:saphire laser with up to 10 μJ available average energy per pulse used for the first-step excitation at 439 nm. Dedicated spectra of the triplet, doublet and singlet multiplets (see Fig. 2a) were acquired varying the average energy per pulse of the excitation step laser (from 0.08 to 1 μJ), to investigate the contribution of the different broadening mechanisms to the line shape. This enabled to separate the contributions of collisional broadening and the natural linewidth (Lorentz) from temperature-associated, laser linewidth and gas-jet divergence broadening (Gauss), see Table 2. For the spectroscopic measurements, the energy per pulse of the first step was set around 0.8 μJ. Two broad-bandwidth ($\sim 4$ GHz) pulsed Ti:sapphire lasers were used simultaneously for the second ionization step at 425 and 435 nm with up to 180 μJ pulse energy (for a detailed description of the laser system see ref. 15). Operated at 10 kHz pulse repetition rate, the lasers with an effective beam spot area of 4 mm illuminated the atomic jet immediately at the nozzle exit. With these ionization conditions, fixed by limitations in the present experimental setup, only a fraction ($\sim 1/14$) of the atoms in the jet could be irradiated, resulting in an overall efficiency of 0.40(13)% and a selectivity of 121(27). The efficiency in the in-gas-cell experiments, where in contrast all atoms are irradiated, is comparable with the in-gas-jet result. This points to collisional de-excitation losses in the ionization process due to the high-pressure environment. The overall in-gas-jet efficiency depends on different factors, namely stopping of the reaction products (100% in this case), transport of the atoms towards the de Laval nozzle (35% diffusion losses, 17% decay losses for the 170 ms $^{215}$Ac), the availability of actinium in its atomic ground state (losses due to non-neutralized ions (>10%) and to molecule formation) and the efficiency of the laser ionization. Increasing the average pulse energy of the first step laser to 9 μJ resulted in a 1.9(2) gain in overall efficiency but the spectral resolution deteriorated. Optimizing the gas-jet formation and shaping the laser beam spot area to a sheet 55 mm long and 3 mm wide would result in a 100% duty cycle[12,30]. In this way, an overall efficiency of >10% and a spectral resolution of $\sim 100$ MHz (FWHM) can be obtained.

**Data evaluation.** For the high-resolution data obtained in the offline and in the gas-jet experiments, the fitting procedure converged only for a particular spin value. This made an unambiguous spin assignment for the isotopes $^{214,215,227}$Ac possible. We obtained the hfs constants, $a$ and $b$, and the centre of gravity of the hfs for each isotope from least-square minimization fits of a 12-peak Voigt profile to the data points. The ratios $a_u/a_l$ and $b_u/b_l$ between the hyperfine constants of the excited (upper) and ground-state (lower) constants were fixed in all the fits to those deduced for $^{227}$Ac from the fully resolved spectra obtained in the offline experiments, neglecting the effect of a possible differential hyperfine anomaly. This effect is generally <1% (see ref. 38) and its influence on the final results can be neglected in comparison with the experimental uncertainties.

We deduced the magnetic dipole moment of the short-lived actinium isotopes from the scaling relation

$$\mu^{exp} = \frac{a^{exp}(6d\,7s\,7p\,^4P^o_{5/2}) \cdot I^{exp}}{a^{227}(6d\,7s\,7p\,^4P^o_{5/2}) \cdot I^{227}} \cdot \mu^{227}, \qquad (1)$$

using our experimental $a$ factors and the calculated value for $^{227}$Ac, and assuming that any possible contribution arising from the presence of hyperfine anomaly between $^{227}$Ac and the lighter actinium isotopes is masked by the large theoretical systematic uncertainty associated with the $\mu^{227}$ value.

In a similar way, we deduced the spectroscopic quadrupole moments using the scaling relation

$$Q^{exp} = \frac{b^{exp}(6d\,7s^2\,^2D_{3/2})}{b^{227}(6d\,7s^2\,^2D_{3/2})} \cdot Q^{227}, \qquad (2)$$

where the calculated quadrupole moment of $^{227}$Ac is used as reference.

The measured isotope shift $\delta\nu^{A,215}$ of two nuclei with mass $A$ and 215 originates from the change in nuclear mass and in nuclear charge distribution and is related to the change in the mean-square charge radius by[39]

$$\delta\langle r^2 \rangle^{A,215} = \langle r^2 \rangle^{215} - \langle r^2 \rangle^A = \frac{1}{F} \cdot \left( \delta\nu^{A,215} - \frac{A-215}{A \cdot 215} \cdot M \right), \qquad (3)$$

with both the mass shift constant $M$ and the field shift constant $F$ obtained by calculations.

**Atomic structure calculations.** The MCDHF method was applied to study the atomic structure of neutral actinium. The experimentally observed levels were identified and subsequently their hyperfine coupling constants $a$ and $b$ were calculated. Furthermore, we determined the isotope-shift parameters $M$ and $F$ for the $^2D_{3/2} \rightarrow {}^4P^o_{5/2}$ transition at 438.58 nm (22,801.1 cm$^{-1}$).

The wave functions were generated using the relativistic atomic structure package GRASP2k[18]. Virtual excitations from a set of reference configurations to a systematically enlarged set of up to five layers of correlation orbitals with angular momenta up to $g$ are used to generate the multiconfiguration basis. Here we took single, double and triple excitations from the valence shells into account. Core-valence correlation was accounted for by performing single excitations from the $6p$-shell together with a second excitation from one of the valence shells. Furthermore, core polarization was also included in our calculations by including single excitations from all core orbitals into the active space.

In a subsequent step, the obtained wave functions were used to extract the isotope-shift parameters and the hyperfine coupling constants $a$ and $b$. By scaling the calculated hfs interaction constants for different energy levels in $^{227}$Ac with those measured in our offline studies, we obtained the values $\mu = 1.07(18)\mu_N$ and $Q = 1.74(10)$ $eb$ for the magnetic dipole and electric quadrupole moments of $^{227}$Ac, respectively.

The mass-shift constant $M$ was obtained by calculating the expectation value of the relativistic recoil operator in the generated eigenfunctions as described in ref. 40. A series of configuration interaction calculations was performed to obtain the isotope-shift between different isotopes for a model nucleus. Subsequently, the field-shift was extracted from these differences in the transition energy. This method, applied to other elements as heavy as polonium[41,42], resembles the experimental approach and we refer to ref. 43 for more details.

The mass-shift constant was found to be $M = 500(180)$ GHz amu, which results in a very small contribution compared with the total isotope shift (see equation (3)) and a field-shift constant of $F = -39(2)$ GHz fm$^{-2}$ was obtained. Several independent calculations were carried out and the resulting differences were used to estimate the uncertainties.

**Nuclear-shell model calculations and additivity rule.** We performed shell-model calculations for ground and excited states in the region northwest of $^{208}$Pb that resulted in the magnetic dipole and electric quadrupole moments for the $N = 123$–$126$ actinium isotopes. We have used the $\pi(h_{9/2}, f_{7/2}, i_{13/2})$, $\nu(f_{5/2}, p_{3/2}, p_{1/2}, i_{13/2})$ model space and the PBPKH interaction[44] from the OXBASH package[45]. The interaction comprises the Kuo–Herling two-body matrix elements for $\pi\pi$ and $\nu\nu$, and for $\pi\nu$ the two-body matrix element from the H7B potential[46]. The large numbers of valence particles and orbits in the model space made it necessary to apply truncation, to make calculations feasible for $Z \geq 87$ and $N < 126$. This in turn implied tuning of the pairing strength, to describe the evolution of occupation and single-particle energies with increasing distance from the core nucleus $^{208}$Pb in a satisfactory way. The degree of truncation was benchmarked using the francium isotopes where experimental data are known. In particular, the high spin $\pi(i_{13/2})$, $\nu(h_{9/2}, f_{7/2}, i_{13/2})$ and the low-spin $\pi(p_{3/2}, p_{1/2})$ were blocked, which leaves a $\pi(h_{9/2}, f_{7/2})$, $\nu(f_{5/2}, p_{3/2}, p_{1/2})$ model space. The effective nucleon charges and $g$-factors used in these calculation that describe the whole region very well are the E2 operators $e_p = 1.5$, $e_n = 0.85$ and the M1 operators $g_s = 0.6 \cdot g_s^{free}$; $g_p^l = 1.115$, $g_n^l = 0$.

The $g$-factors of the ground state $(9/2^-)$ and high-spin $(17/2^-$ and $21/2^-)$ isomers that are based on a proton $h_{9/2}$ configuration in $^{215}$Ac have been calculated to be 0.912, 0.925 and 0.969, respectively. Only the value for both isomers agree with the experimental value 0.920(19)[24].

Semi-magic nuclei can be described in a seniority scheme[47]. Hereby, a $j^n$ configuration with spin $I$ is additionally labelled by the seniority $\nu$, which counts the number of unpaired nucleons. In this framework, the E2 matrix element is given by $<j^n, \nu I \| E2^n, \nu I> \sim Q(I) = [2j+1-2n]/[2j+1-\nu] <j\|E2>$ and is linear in $n$ and changes sign in the half-filled orbit. The continuous linear trend in the ground-state quadrupole moments of the $N = 126$ isotones (seniority $\nu = 1$) from $Z = 83$ to 89 classifies these nuclei as semi-magic.

In the analysis of the odd–odd nuclei we compare our results for the magnetic moments with those provided by the additivity rule[48] for different spin values. For both isotopes $^{212,214}$Ac the experimental values are predicted by simple coupling rules in combination with the suitable nucleon configuration involving only the

experimental magnetic moments of the neighbouring odd-$A$ nuclei, namely $^{211,213}$Ra (refs 22,23) for the neutrons and the in-gas-jet result of $^{215}$Ac for the protons (see Table 2).

**Data availability.** All the relevant data supporting the findings of these studies are available from the corresponding author upon request.

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

## Acknowledgements

We thank Nancy Postiau and the operators of the CRC Louvain-la-Neuve for the technical support and the preparation of the stable neon beams. This work was supported by FWO-Vlaanderen (Belgium), by GOA/2010/010 (BOF KU Leuven), by the IAP Belgian Science Policy (BriX network P7/12) and by a Grant from the European Research Council (ERC-2011-AdG-291561-HELIOS). S.S. acknowledges a Ph.D. Grant of the Belgian Agency for Innovation by Science and Technology (IWT). L.P.G. acknowledges FWO-Vlaanderen (Belgium) as an FWO Pegasus Marie Curie Fellow. S.F. and R.B. acknowledge the support by the German Ministry for Education and Research (BMBF) under contract No. 05P15SJCIA.

## Author contributions

M.H., Yu.K., P.V.D. and R.F. conceived the on-line experiments. K.W., I.D.M., N.L. and F.L. provided the extra experimental equipment. Yu.K., S. Rae. and R.F. set up the broadband laser system. S. Rae. commissioned and operated the narrowband laser system with the help of T.K., R.H., P.N., N.L. and K.W. Yu.K., S.S. and R.F. set up the gas cell and optimized the ion transport. L.G., S.S., P.V.d.B. and R.F. set up the detection system. B.B., P.C., R.d.G., P.D., X.F., S.F., L.P.G., L.G., C.G., R.H., L.H., M.H., T.K., Yu.K., M. Laa., I.D.M., Y.M., E.M., P.N., J.P., S. Rae, S. Rot, H.S., S.S., V.S., J.C.T., E.T., C.V.B., P.V.d.B., P.V.D., K.W., A.Z. and R.F. performed the on-line experiments. M. Loi. coordinated the operation of the cyclotron and set-up the neon beams. S. Rae., V.S., T.K., R.H., P.N., K.W., C.G. and R.F. performed the offline measurements. A.B., S. Rae., C.G., P.V.D., Yu.K., W.G., M.H., L.P.G., S.S. and R.F. analysed and interpreted the data. R.B. and S.F. performed the MCDF calculations. H.G. performed the shell-model calculations. M.H., P.V.D., Yu.K. and R.F. supervised the project. The paper was written by R.F., M.H. and P.V.D. with input from A.B., S. Rae. Yu.K., L.P.G., L.G., B.B., I.M., M.B., X.F., H.G., S.F. and R.B.

## Additional information

**Competing financial interests:** The authors declare no competing financial interests.

