## [Peer Review File · Nature Communications]

Reviewers' comments:

Reviewer #1 (Remarks to the Author):

This paper describes a new method for performing high resolution laser spectroscopy measurements on isotopes of practically any element that can be created with intensities as low as 1 every 10 seconds, as long as the lifetime is longer than 100msec. It has been successfully demonstrated with a specific case of isotopes of Actinium.

The key new step was to do the laser ionization in a gas jet that reduces much of the temperature and pressure broadening that occurs in gas cells. The improved resolution allowed the extraction of small splittings that were not observable in gas cell work, that were necessary to extract the Q moment of the ground state.

The existing apparatus had some limitations that the authors indicate can be improved in future devices, in particular to improve the overlap with all of the atoms in the gas jet which is not currently matched well to the laser depletion rate. An improved device holds great promise for extending nuclear spectroscopic measurements to the heaviest elements that have had limited study so far. It will also be of interest to groups working at radioactive beam facilities to advance the study of nuclear properties far from stability.

The paper is well written and describes the measurements and calculations in sufficient detail to understand the work, and the paper is adequately referenced.

Reviewer #2 (Remarks to the Author):

This paper is well written and well organized. The authors have done a good job of presenting their work and clearly spent some effort on the editing, which has resulted in a paper that is easy to read.

The paper describes the development of a novel technique using resonant laser ionization to ionize short-lived, high-mass radioisotopes. Using this high-resolution laser spectroscopy technique to selectively ionize actinium isotopes in the ground-state or the isomeric-state, nuclear properties such as magnetic-dipole moments and electric-quadrupole moments, were deduced. Improvements in the laser-atom overlap region resulted in a much-improved capability to select specific atomic transitions in the hyperfine structure with very high resolution. The authors are quite clear about the limitations of this particular experiment and provide a realistic estimate of the eventual capability of this technique.

I have a few comments and questions, but no major changes are suggested.

1. In the second paragraph, the provisional name for the element with $Z=118$ is used as if it is final (maybe it will be by the time this paper is in print).
2. On the first page, the acronym, IGLIS, is defined as "In-Gas ..." but in the title for Table 1 it seems to refer to "In-Gas-Jet ...". Does IGLIS refer to the more generic technique or to this particular setup?
3. In Figure 1, the scale for the pressure indicates that the color of the gas cell chamber (i.e. outside of the gas cell) should be dark blue. Maybe this could be changed to just include the pressures down to what is seen in the RF ion guide. Also, in Figure 1, what is the purpose of the plates (labeled "+" and "_") just before the de Laval nozzle.
4. In Figure 2, the "-" in the x-axis label should be clearly defined as a minus sign and not just a dash. It should be immediately clear to the reader that the axis values are the difference in frequency (GHz) between the actual frequency and the value given in the label.
5. In the last note for Table 1, the selectivity is related to the ions "stopped in the buffer gas". Is there a big difference in the number of ions stopped in the buffer gas and what is produced in the reaction

of the beam with the Au target? Does this number refer only to "ions" in the buffer gas or does it include all atoms of interest? Bottom line: is there a particular reason that you used ions in the buffer gas instead of what was produced in the reaction?

I can recommend this paper for publication without reservation. The comments above are suggestions for improvements, but should not be construed as required changes to correct errors.

Reviewer #3 (Remarks to the Author):

Referee report for NCOMMS-16-20306-T "Resonance ionisation in a Supersonic Gas Jet: towards high-resolution laser spectroscopy of the heaviest elements" by R. Ferrer, et al.

This paper discusses adapting resonance laser ionization techniques to investigate atomic and nuclear properties of radionuclides in a supersonic gas jet. This is novel and enables investigation of radionuclides that are produced on-line and might have half-lives that are too short for traditional methods of high-resolution laser spectroscopy. Improving the efficiency will enable measurement of radionuclides produced with low cross-section. This technique could be applied for study of isomeric states in addition to ground state properties of nuclei, like size, shape, spin and electromagnetic multipole moments. Additionally, the technique is chemically independent, allowing access to traditionally difficult elements to measure such as refractory elements. This paper described the adapted technique and examined Ac isotopes near the N=126 shell closure to demonstrate the measurements that are possible. Long-lived ^{227}Ac was measured to demonstrate the resonant laser ionization scheme and compare with previous measurements, before ^{212}Ac - ^{215}Ac isotopes were produced and the in-beam measurements were made. The performance of the system on ^{215}Ac (half-life = 170 ms) was impressive.

This paper is very clearly written – in fact, one of the best paper's I've ever reviewed for clarity. The paper includes sufficient detail that the work could be reproduced or systems could be constructed at other laboratories. In short, a very well-written paper with strong conclusions that is of interest to a wider field than just laser spectroscopists. This technique enables the investigation of numerous other isotopes, especially in the actinide and trans-actinide region, that in the past were inaccessible. The scientific community stands to gain an enormous amount of information about these isotopes that will impact efforts to understand the structure of the nucleus as well as the formation of the heavy and superheavy elements. This technique now needs to be applied to other isotopes. These measurements will undoubtedly influence nuclear structure calculations and model development in this region of the chart of nuclides.

The paper is so well-written that I only have a few minor comments/questions:

1) In Figure 1 – I understand the desirability to not have the gas-jet and laser beam co-linear in order to not complicate interpretation due to the fraction ionized in the gas cell perhaps, but couldn't the geometry of the interaction region be improved for better/longer overlap? A series of mirrors to bounce the laser beams back and forth across the gas jet? Angling the beams? Seems to me that this would improve the efficiency of the technique. Later on page 9, there is a paragraph devoted to discussing the spatial limitations and "projected improvements" are mentioned but not discussed. This seems to be a place where a sentence might be added to discuss the geometrical improvements in a little more detail. It is good that improvements are in the works, but it is also difficult for the reader to judge if 10% efficiency is really achievable.

2) Page 3 – line 87 – suited \diamond suitable – "... including isomeric beams, and is suitable for further studies." This paper did not demonstrate measurement of an isomer – has there been any try?

3) Table 1 – do the projected gas jet numbers include improved geometry? What is the main improvement? In other words, what are the “working conditions”?

I heartily recommend publication of this work immediately in Nature Communications following consideration of the minor points above.

Point-by-point response to reviewers comments/suggestions

Reviewers' comments:

Reviewer #1 (Remarks to the Author):

This paper describes a new method for performing high resolution laser spectroscopy measurements on isotopes of practically any element that can be created with intensities as low as 1 every 10 seconds, as long as the lifetime is longer than 100msec. It has been successfully demonstrated with a specific case of isotopes of Actinium.

The key new step was to do the laser ionization in a gas jet that reduces much of the temperature and pressure broadening that occurs in gas cells. The improved resolution allowed the extraction of small splittings that were not observable in gas cell work, that were necessary to extract the Q moment of the ground state.

The existing apparatus had some limitations that the authors indicate can be improved in future devices, in particular to improve the overlap with all of the atoms in the gas jet which is not currently matched well to the laser depletion rate. An improved device holds great promise for extending nuclear spectroscopic measurements to the heaviest elements that have had limited study so far. It will also be of interest to groups working at radioactive beam facilities to advance the study of nuclear properties far from stability.

The paper is well written and describes the measurements and calculations in sufficient detail to understand the work, and the paper is adequately referenced.

Reviewer #2 (Remarks to the Author):

This paper is well written and well organized. The authors have done a good job of presenting their work and clearly spent some effort on the editing, which has resulted in a paper that is easy to read.

The paper describes the development of a novel technique using resonant laser ionization to ionize short-lived, high-mass radioisotopes. Using this high-resolution laser spectroscopy technique to selectively ionize actinium isotopes in the ground-state or the isomeric-state, nuclear properties such as magnetic-dipole moments and electric-quadrupole moments, were deduced. Improvements in the laser-atom overlap region resulted in a much-improved capability to select specific atomic transitions in the hyperfine structure with very high resolution. The authors are quite clear about the limitations of this particular experiment and provide a realistic estimate of the eventual capability of this technique.

I have a few comments and questions, but no major changes are suggested.

1. In the second paragraph, the provisional name for the element with $Z=118$ is used as if it is final (maybe it will be by the time this paper is in print).

Authors: To prevent from a possible misunderstanding we have decided to avoid using the provisional name and have used instead "Element $Z=118$ is thus far..."

2. On the first page, the acronym, IGLIS, is defined as "In-Gas ..." but in the title for Table 1 it seems to refer to "In-Gas-Jet ...". Does IGLIS refer to the more generic technique or to this particular setup?

Authors: Indeed, IGLIS refers to the basic and generic technique that can be performed by applying the in-gas-cell or in-gas-jet methods. Consequently, in Table 1 IGLIS refers to the current (in-gas-cell and in-gas-jet) and the projected (in-gas-jet) performance of both methods. We have adopted this terminology and it is used throughout the manuscript (changes are highlighted in the text).

3. In Figure 1, the scale for the pressure indicates that the color of the gas cell chamber (i.e. outside of the gas cell) should be dark blue. Maybe this could be changed to just include the pressures down to what is seen in the RF ion guide.

Authors: We perfectly understand this remark and are aware of the inconsistent color code in the picture. The problem is that by adding a blue background to the volume of the gas cell chamber we would obtain a very dark picture making the visualization of other elements in it more difficult. For the sake of clarity we have delimited the gas cell chamber volume with a dashed-line box and have indicated in the figure caption the value of the background pressure in such a volume.

Also, in Figure 1, what is the purpose of the plates (labeled "+" and "-") just before the de Laval nozzle.

Authors: These plates are the electrodes used to create an electric field and collect the survival ions. We have adapted the figure caption with the sentence "...to collect the remaining ions by applying a DC voltage on a pair of electrodes indicated by a + and a - sign."

4. In Figure 2, the "-" in the x-axis label should be clearly defined as a minus sign and not just a dash. It should be immediately clear to the reader that the axis values are the difference in frequency (GHz) between the actual frequency and the value given in the label.

Authors: We have changed the abbreviation 'Freq.' by the Greek letter 'nu' in the labels for the x-axis and have indicated in the figure caption that the axis indicates the frequency detuning with respect to the value of the center of gravity.

5. In the last note for Table 1, the selectivity is related to the ions "stopped in the buffer gas". Is there a big difference in the number of ions stopped in the buffer gas and what is produced in the reaction of the beam with the Au target?

Authors: About 40% of the total ^{215}Ac nuclei produced in the reaction of ^{22}Ne with the gold target of 1.931 mg cm^{-2} are stopped within the target and the rest leave the target and are stopped in the buffer gas.

Does this number refer only to "ions" in the buffer gas or does it include all atoms of interest?

Authors: That number refers therefore to the ^{215}Ac nuclei that leave the target and are stopped in the buffer gas. The note in Table 1 has been rephrased for clarity as "Ratio between the ^{215}Ac ions entering the mass separator to the ^{215}Ac nuclei stopped in the buffer gas."

Bottom line: is there a particular reason that you used ions in the buffer gas instead of what was produced in the reaction?

Authors: The reason is that not all nuclei produced in the reaction leave the target. Thus, in order to correctly quantify the total efficiency of the IGLIS technique, irrespective of the used reaction, one has to consider only those nuclei stopped in the buffer gas.

I can recommend this paper for publication without reservation. The comments above are suggestions for improvements, but should not be construed as required changes to correct errors.

Reviewer #3 (Remarks to the Author):

Referee report for NCOMMS-16-20306-T “Resonance ionisation in a Supersonic Gas Jet: towards high-resolution laser spectroscopy of the heaviest elements” by R. Ferrer, et al.

This paper discusses adapting resonance laser ionization techniques to investigate atomic and nuclear properties of radionuclides in a supersonic gas jet. This is novel and enables investigation of radionuclides that are produced on-line and might have half-lives that are too short for traditional methods of high-resolution laser spectroscopy. Improving the efficiency will enable measurement of radionuclides produced with low cross-section. This technique could be applied for study of isomeric states in addition to ground state properties of nuclei, like size, shape, spin and electromagnetic multipole moments. Additionally, the technique is chemically independent, allowing access to traditionally difficult elements to measure such as refractory elements. This paper described the adapted technique and examined Ac isotopes near the N=126 shell closure to demonstrate the measurements that are possible. Long-lived ^{227}Ac was measured to demonstrate the resonant laser ionization scheme and compare with previous measurements, before ^{212}Ac - ^{215}Ac isotopes were produced and the in-beam measurements were made. The performance of the system on ^{215}Ac (half-life = 170 ms) was impressive.

This paper is very clearly written – in fact, one of the best paper’s I’ve ever reviewed for clarity. The paper includes sufficient detail that the work could be reproduced or systems could be constructed at other laboratories. In short, a very well-written paper with strong conclusions that is of interest to a wider field than just laser spectroscopists. This technique enables the investigation of numerous other isotopes, especially in the actinide and trans-actinide region, that in the past were inaccessible. The scientific community stands to gain an enormous amount of information about these isotopes that will impact efforts to understand the structure of the nucleus as well as the formation of the heavy and superheavy elements. This technique now needs to be applied to other isotopes. These measurements will undoubtedly influence nuclear structure calculations and model development in this region of the chart of nuclides.

The paper is so well-written that I only have a few minor comments/questions:

1) In Figure 1 – I understand the desirability to not have the gas-jet and laser beam co-linear in order to not complicate interpretation due to the fraction ionized in the gas cell perhaps, but couldn’t the geometry of the interaction region be improved for better/longer overlap? A series of mirrors to

bounce the laser beams back and forth across the gas jet? Angling the beams? Seems to me that this would improve the efficiency of the technique.

Authors: That is correct and is envisaged for the optimal setup. A multi-pass mirror system could be used to increase the efficiency while keeping an optimum spectral resolution, as in this way, one could avoid additional power broadening. This clarification has been added in the main text (see next remark).

Later on page 9, there is a paragraph devoted to discussing the spatial limitations and “projected improvements” are mentioned but not discussed. This seems to be a place where a sentence might be added to discuss the geometrical improvements in a little more detail. It is good that improvements are in the works, but it is also difficult for the reader to judge if 10% efficiency is really achievable.

Authors: In the manuscript we show that in case of an optimized duty cycle during the reported measurements we could have reached a total efficiency of about 6%. The projected 10% is based on the fact that in the future IGLIS setups we will use a new gas cell design that has been optimized in dedicated flow-dynamic simulations, unlike the gas cell used in the reported experiments, for diffusion losses using the professional software package COMSOL. A summary of these simulations has been recently reported in ref [32]. The results from such simulations, not yet confronted with experimental data, point to an increase of the extraction efficiency with respect to the former gas cell design that might well reach a factor of two.

Following the referee’s advice we have modified the aforementioned paragraph that now reads: “.....the projected performance of the technique under optimal conditions, which also include a new gas-cell design with better transport and extraction characteristics [32]. Thus, assuming a duty cycle adapted to the supersonic expansion through a de Laval nozzle with a modest Mach number 10, tuning the background pressure for an extended pencil-like jet and optimizing the laser interaction zone, through a multi-pass mirror system, a final spectral”

2) Page 3 – line 87 – suited \diamond suitable – “... including isomeric beams, and is suitable for further studies.”

Authors: We have changed the word ‘suited’ by ‘suitable’ but notice that “suitable for further studies” refers to “pure ion beams” and not to “The In-Gas-Jet laser ionization and spectroscopy method”.

Therefore, the new sentence reads: “This In-Gas-Jet laser ionisation and spectroscopy method enables the production of extremely pure ion beams, including isomeric beams, which are suitable for further studies.

This paper did not demonstrate measurement of an isomer – has there been any try?

Unfortunately there were no isomers in the actinium isotopes, with the exception of those in ^{215}Ac with ns half-life, that we could measure using the in-gas-jet method. In a former publication (R. Ferrer et al. PLB 728 (2014) 191) we used the in-gas-cell method to perform laser spectroscopy on the isomers in $^{101,99}\text{Ag}$ that could be perfectly separated from their ground states despite of the inability to resolve their hyperfine structure owing mainly to collisional broadening. Using the in-gas-jet method in future experiments, we will be able, for instance, to produce high purity beams of

these isomers and even to resolve, at least partially, their very collapsed hyperfine structure.

3) Table 1 – do the projected gas jet numbers include improved geometry? What is the main improvement? In other words, what are the “working conditions”?

Authors: Yes, they do. See comments to questions in 1).

I heartily recommend publication of this work immediately in Nature Communications following consideration of the minor points above.

To the Editor:

In addition to amending the changes pointed out by the referees we have also corrected the statistical uncertainties that were erroneously considered as only 1 sigma rather than 2 sigma in the isotope shift values $\delta\nu^{A,215}$ of the gas cell data given in Table 2. The statistical errors in the deduced mean-square charge radii of these isotopes have correspondingly been modified. The modified values are highlighted in the pdf version. Furthermore, other changes implemented in the text to comply with the format requirements of NComms. are also highlighted in the pdf file.

REVIEWERS' COMMENTS:

Reviewer #2 (Remarks to the Author):

The authors have done a good job of addressing the issues raised by the reviewers. The paper looks very good and I recommend publication as is.

Reviewer #3 (Remarks to the Author):

The authors have addressed all of the concerns of the referees. I recommend publication.

REVIEWERS' COMMENTS:

Reviewer #2 (Remarks to the Author):

The authors have done a good job of addressing the issues raised by the reviewers. The paper looks very good and I recommend publication as is.

Reviewer #3 (Remarks to the Author):

The authors have addressed all of the concerns of the referees. I recommend publication.